# Effects of a Novel Applet-Based Personalized Dietary Intervention on Dietary Intakes: A Randomized Controlled Trial in a Real-World Scenario

**DOI:** 10.3390/nu16040565

**Published:** 2024-02-19

**Authors:** Hongwei Liu, Jingyuan Feng, Zehuan Shi, Jin Su, Jing Sun, Fan Wu, Zhenni Zhu

**Affiliations:** 1School of Public Health, Fudan University, Shanghai 200032, China; hwliu23@m.fudan.edu.cn (H.L.); jingyuanfeng21@m.fudan.edu.cn (J.F.); 2Division of Health Risk Factors Monitoring and Control, Shanghai Municipal Center for Disease Control and Prevention, Shanghai 200336, China; shizehuan@scdc.sh.cn (Z.S.); sujin@scdc.sh.cn (J.S.); 3National Institute for Nutrition and Health, Chinese Center for Disease Control and Prevention, Beijing 100050, China; sunjing@ninh.chinacdc.cn

**Keywords:** personalized nutrition, behavioral nutrition interventions, dietary intakes, randomized control trial (RCT), nutrition translation

## Abstract

The objective of this study was to assess the feasibility and effectiveness of a novel WeChat applet-based personalized dietary intervention aimed at promoting healthier dietary intakes. A two-arm parallel, randomized, controlled trial was conducted in a real-world scenario and involved a total of 153 participants (the intervention group, *n* = 76; the control group, *n* = 77), lasting for 4 months in Shanghai, China. The intervention group had access to visualized nutrition evaluations through the applet during workday lunch time, while the control group received no interventions. A total of 3413 lunch dietary intake records were captured through the applet. Linear mixed models were utilized to assess the intervention effects over time. At baseline, the participants’ lunchtime dietary intakes were characterized by insufficient consumption of plant foods (86.9% of the participants) and excessive intake of animal foods (79.7% of the participants). Following the commencement of the intervention, the intervention group showed a significant decrease in the animal/plant food ratio (β = −0.03/week, *p* = 0.024) and the consumption of livestock and poultry meat (β = −1.80 g/week, *p* = 0.035), as well as a borderline significant increase in the consumption of vegetables and fruits (β = 3.22 g/week, *p* = 0.055) and plant foods (β = 3.26 g/week, *p* = 0.057) over time at lunch compared to the control group. The applet-based personalized dietary intervention was feasible and effective in improving dietary intakes and, consequently, possibly may manage body weight issues in real-world scenarios.

## 1. Introduction

Unhealthy dietary intakes (i.e., excessive consumption of sodium, fat, sugar, red meat, and processed meat; low intake of fruits and vegetables) are a major contributing factor to the growing incidence of unhealthy states and various medical conditions [1,2]. The dramatic development of the fast-food and packaged foods industry has also indulged people’s choices in unhealthy foods. As proven in a large body of evidence, interventions against unhealthy dietary intakes and subsequent improvements in dietary intakes are associated with numerous health benefits, such as better metabolic conditions, fewer cardiovascular diseases, and fewer cancers [3,4,5]. There is a need to translate these intervention programs into real-world practices to reduce the burden of disease associated with unhealthy dietary intakes.

Although dietary intervention conclusively plays a positive role in keeping fit and further preventing diseases, its widespread implementation has been impeded due to several limitations. In-person coaching is resource-intensive, time-consuming, and costly, requiring many professionals; therefore, its scalability and outreach to a large number of people in need is limited [6]. Among large-scale dietary intervention programs on community or social levels, low uptake, lack of individualized measures, and high attrition are widespread [7,8]. These limitations hinder the potential efficacy of these interventions in real-world settings.

Previous studies have partially shown that, compared to traditional dietary interventions, personalized dietary interventions based on mobile apps are more cost-effective, available, and visually appealing [9,10].

The main objective of this study was to evaluate the effectiveness and feasibility of a novel personalized dietary intervention provided by an applet. The intervention was delivered in two stages (pre-meal: “traffic light” illustrations of dish nutrition evaluation; post-meal: personalized nutrition report of the meal consumed) and designed to assist participants in consuming healthier foods.

## 2. Materials and Methods

### 2.1. Study Design

A two-arm parallel, randomized, controlled trial under a real-world scenario was conducted to test the effectiveness of a personalized dietary intervention provided by an applet. The applet was a WeChat (like Facebook in the Western world) mini-program, a type of lightweight application that could run on the WeChat platform without the need for users to download and install it [11]. The applet was designed for buffet-style canteens where each dish was prepared by fixed recipes and served in single-portion sizes [12]. The participants were enrolled sequentially until the sample size reached the requirement. This study was conducted in a pilot company in Shanghai, China with 3000 employees, and food was prepared by the central kitchen in the staff canteen. All participants were free to follow their daily routines and enjoy unrestricted meals at will. More details on the study design and the applet had been previously published as a protocol [12].

Dietary data were collected each time the participants ordered lunches through the applet. Anthropometric measurements were taken at baseline and subsequent monthly intervals throughout the study duration. Additionally, two questionnaire surveys were conducted at baseline and approximately one month after the study initiation.

### 2.2. Participants

Recruitment was conducted from September 2022 to November 2022, with the study beginning in September 2022 and lasting for 4 months. The inclusion criteria were as follows: aged more than 18 years; apparently healthy; promising to have lunch at the staff canteen during the study period; agreeing to record food consumption of each meal on the applet. The exclusion criteria were as follows: planning to change dietary and physical activity habits in the next 4 months; going on special diets (i.e., not having lunch, on diet, etc.). The study aimed to recruit 70 participants in each arm to detect an assumptive intervention effect size of 0.25 in their animal/plant food ratio over 4 months, with an SD change in animal/plant food ratio of 0.41 from the Shanghai Diet and Health Survey [13], a 90% statistical power, a 5% significance level, and a 15% dropout rate [12].

A total of 153 of 177 enrolled participants were randomly allocated either to the intervention group (*n* = 76) or the control group (*n* = 77). During the follow-up stage, all participants provided at least one dietary record, totaling 3413 records. Approximately half of participants (*n* = 65) did not participate in follow-up anthropometric measurements due to the impact of the COVID-19 epidemic and the quarantine policy, and 20 participants in the intervention group did not complete the follow-up questionnaire. Additionally, a subset of participants changed their exercise routine (*n* = 15) or adjusted their lunch dietary intake as a proportion of their total daily dietary intake (*n* = 5). These participants were included in the main analysis to assess the intervention effect but were excluded from the sensitivity analysis to evaluate its robustness (Figure 1).

### 2.3. Randomization and Blinding

The participants were randomized to either the intervention or control group in an allocation ratio of 1:1, using stratified randomization by sex and age. The trial was conducted as a double-blind study. The researchers were not blinded to allocation due to the nature of the intervention. However, the field investigators and participants were blinded throughout the study to ensure allocation concealment.

### 2.4. Run-In Phase and Baseline Assessment

Eligible participants were invited to undergo a 1 week run-in period to learn how to use the applet for ordering dishes and entering leftover proportions during their weekday lunchtime in the staff canteen. During this phase, participants in both groups were unable to access the dietary nutrition evaluation functionality of the applet and could only perform the above-mentioned operations. Baseline data collected during this phase included an online questionnaire, anthropometric measurements, and dietary data recorded using the applet.

### 2.5. Interventions

Immediately after the 1 week run-in phase, the participants in the intervention group were given access to the dietary nutrition evaluation functionality of the applet, including pre-meal “traffic light” illustrations of dishes (Figure 2b) and post-meal personalized nutrition reports (Figure 2c), while those in the control group were not, similar to the run-in phase. Both groups were instructed to continue using the applet to order and then consume their weekday lunch in the staff canteen. In the canteen, each dish was prepared by fixed recipes and then divided into equal weight single-serving portions for diners to choose from (Figure 2a). By integrating the pre-collected recipe dataset of all the dishes supplied in the canteen with the publicly accessible Chinese food composition database at the applet’s backend, the applet’s dietary nutrition evaluation functionality was real-time realized and visualized as the participants ordered dishes via the applet in the canteen [12,14]. All participants in both groups chose and consumed meals of their own free will.

#### 2.5.1. Pre-Meal Intervention: Figure 2b

In the food-ordering interface, the participants browsed and selected dishes labeled with colored dots adjacent to their names. The dots were color-coded based on the fat, sodium, and sugar levels of dishes using a “traffic light” approach (green = reaching the dietary recommendations, yellow = between the recommendations and average intakes of the Chinese population [15], red = above the upper limit of intakes). These color codes were automatically calculated at the applet’s backend [12]. The cutoffs of nutrient contents are listed in Table 1.

#### 2.5.2. Post-Meal Intervention: Figure 2c

A real-time personalized nutrition report on each meal was visually displayed after dishes were ordered and leftover proportions were entered.

In the nutrition report module, actual intakes of protein, fat, and carbohydrates were first shown using a colored circle: green for reaching the recommended range, yellow for slight deviations, and red for serious deviations [16]. Then, for food groups (e.g., cereals and tubers, vegetables, red meat, and poultry) and other nutrients, actual intakes were displayed as dots and recommended intakes as bars. The positional relationship between dots and bars visually showcased whether actual intakes aligned with recommended intakes and the extent of deviation from the recommendations. These recommended intakes were assessed based on the individual’s sex, age, and estimated energy requirement (EER). EER was calculated using basic energy expenditure (BEE) multiplied by PAL [17]. BEE was estimated by a Schofield equation based on age, sex and body weight [18]. PAL was categorized into 1.5 for light, 1.75 for moderate, and 2.0 for vigorous physical activity.

### 2.6. Measures and Follow-Up

#### 2.6.1. Online Questionnaire Survey

The baseline questionnaire survey collected data on age, sex, smoking status (non-smoker, ex-smoker, and current smoker), alcohol consumption (lifetime abstainer, non-heavy drinker, and heavy drinker), physical activity level (low, moderate, and vigorous), and intentional physical exercise (yes or no). Details of baseline questionnaire assessments have been reported elsewhere [12]. In the post-intervention questionnaire survey, two additional questions were included to assess the ease and understandability of the applet in the intervention group:

Applet usability: The question “Can you operate this applet?” was used to assess the ease of use of the applet. The response options included “Completely mastering”, “Partly mastering”, and “Not mastering”.

Understandability of information provided by the applet: This question “Can you understand the knowledge from the applet?” was used to evaluate the users’ comprehension of dietary evaluations presented in the applet. The response options were “Fully understanding”, “Partly understanding”, and “Not understanding”.

#### 2.6.2. Dietary Records

The primary outcome for the intervention effect was the participants’ lunchtime dietary intake each weekday, including intakes of food groups, energy, and nutrients recorded by the applet. In the canteen, each dish was prepared by fixed recipes and then divided into equal weight single-serving for diners to choose from. Each single original material of the dishes, as well as the condiments applied, were weighed and uploaded to the applet’s backend. After participants ordered dishes and entered the leftover proportions of each dish using the applet during weekday lunchtime, a detailed dietary report was automatically calculated and recorded.

Prior to the launch of the applet, we implemented a process to compile a nutrition dataset for all the dishes supplied in the canteen. First, the raw weights of the materials and ingredients (including condiments) and their edible proportion included in each dish was measured, and then the cooked weight of each dish was further measured. Second, the single-serving cooked weight of each dish was measured. These processes were carried out by field investigators with expertise in nutrition, allowing us to construct an accurate recipe dataset for all the dishes provided in the canteen. Finally, by connecting the recipe dataset to the publicly accessible Chinese food composition database in the applet’s backend, the food groups, nutrients, and energy of each single-serving dish were automatically calculated to compile the nutrition dataset for all the dishes supplied in the canteen.

After a participant ordered dishes and entered the leftover proportion, the intakes of food groups, nutrients, and energy from each single-serving dish consumed by the participant were first calculated by multiplying by non-leftover proportions based on the above nutrition dataset. The overall intake of food groups, nutrients, and energy for the participant’s meal was then generated by summing up the intakes from different single-serving dishes.

Food groups were classified as cereals and tubers (grains, potatoes, and tubers); vegetables (excluding legumes); fruits (including citrus); livestock and poultry meat; eggs; aquatic products; dairy; nuts; soybeans and soybean products; cooking oil; salt; and sugar. Nutrients included protein, fat; carbohydrate; cholesterol; sodium; calcium; iron; zinc; and vitamin C. Intakes of plant foods (cereals and tubers, vegetables, fruits, soybeans and soybean products, and nuts), animal foods (livestock and poultry meat, eggs, aquatic products, and dairy), the animal/plant food ratio, and the percentage of energy intake from fat were further calculated. Given that the intake of fruits was quite low at lunch, we calculated the intake of vegetables and fruits by the intake of vegetables plus the intake of fruits.

Considering fluctuations in dietary intakes, we calculated weekly average intakes to mitigate the potential impact of extreme values of dietary intake on the analysis.

#### 2.6.3. Anthropometric Measurements

Anthropometric indictors (body weight, body mass index (BMI), body composition, and blood pressure) were the secondary outcomes, and these measurements were obtained at baseline and repeated monthly during the follow-up period. Body composition included percentage of body fat, percentage of torso fat, and visceral fat index, measured by a bioelectrical impedance analysis (RD-545, TANITA, Tokyo, Japan). Body weight was also measured by the above analyzer to the nearest 0.1 kg. Body height was measured by a portable stadiometer with an accuracy of 0.1 cm (WEF111, SENSSUN, Zhongshan, China). BMI was calculated based on the following formula: weight/height^2^ (kg/m^2^). Resting blood pressure, including systolic blood pressure and diastolic blood pressure, was measured 2 times by an electronic sphygmomanometer (U30, OMRON, Kyoto, Japan) following a 5 min rest, with a 10 min interval between each measurement.

### 2.7. Statistical Analysis

Baseline characteristics of the participants in both groups and for all participants were presented as the mean (SD) for continuous variables and as frequency (percentage) for categorical variables. Differences in demographic characteristics between the included and excluded participants were evaluated using independent samples t-tests for continuous variables and chi-square tests for categorical variables.

To assess potential bias from the canteen meal service on the intervention effect, we applied the Mann–Kendall test to examine the temporal trend in the animal/plant food ratio of the canteen daily lunchtime menu during the study period.

Mixed model analysis can tolerate missing data at one or more assessments through maximum likelihood estimation and take the interdependency of measures into account [19]. Thus, given repeated measurements for each participant and missing data points in the current study, linear mixed models with restricted maximum likelihood were conducted to analyze the intervention effects on primary and secondary outcomes over time. Time point, group (intervention and control), and a two-way interaction effect between time point and group were included as fixed factors. Random effects were included for repeated measurements for the same participant. A significant interaction item indicated overall trend differences between the groups.

Three models were run to verify the stability of the intervention effect. Model 1 adjusted the baseline value of the dependent variable and enrollment sequence (September, October, and November). Model 2 further adjusted age, sex, smoking status, alcohol consumption, physical activity level, and intentional physical exercise. Additionally, in the analysis of the intervention effect on the primary outcome, Model 2 also adjusted baseline BMI. Based on Model 2, Model 3 further excluded the participants who changed the exercise routine or the lunch dietary intake as a proportion of total daily dietary intake during the follow-up period as a sensitivity analysis.

Additional analysis focused on evaluating the ease and understandability of the applet, presented as a percentage.

All data analyses were performed using R (version 4.2.3). The “nlme” package was used for conducting linear mixed effects models in the study. Statistical significance was set at a two-tailed *p* < 0.05.

## 3. Results

### 3.1. Participants’ Characteristics

As indicated in Table 2, the baseline demographic characteristics and baseline values of outcome variables were comparable between both groups, except for BMI. The intervention group demonstrated a significantly higher mean BMI (mean = 23.9 kg/m^2^, SD = 3.6) than the control group (mean = 22.6 kg/m^2^, SD = 3.8). The mean age of all participants was 32.7 years (SD, 7.5), with 63.4% being female. Appendix A reveals that, compared to the included participants, the excluded participants showed a higher proportion of males, smokers, and drinkers. Otherwise, there were no differences in other demographic characteristics between those who were (*n* = 153) and were not (*n* = 24) included (Appendix A).

Figure 3 illustrated the distributions of baseline lunchtime dietary intakes (plant foods, vegetables and fruits, animal foods, and livestock and poultry meat), BMI, and percentage of body fat among the participants included in the main analysis. The majority of participants exhibited inadequate consumption of plant foods (86.9%) as well as vegetables and fruits (79.7%), and they exhibited excessive intakes of animal foods (79.7%) and livestock and poultry meat (79.7%). Regarding BMI, 28.4% of the participants were overweight, with 11.4% classified as obese. When considering the percentage of body fat, a higher proportion of the participants (35.2%) were classified as overweight, and 36.4% were categorized as obese.

### 3.2. Canteen Meal Supply

During the study period, the median number of dishes provided in the canteen each weekday lunch was 14 (interquartile range (IQR): 14–14), including 5 (5–6) green-coded, 7 yellow-coded (5–8), and 2 (1–3) red-coded dishes. As presented in Figure 4, there was no significant trend over time in the animal/plant food ratio (*p* = 0.134) during the intervention duration.

### 3.3. Effects of the Intervention on Dietary Intakes

Table 3 showed the between-group trend differences in food group consumption and nutrient intake over time using several linear mixed models. The results were mostly similar across the two models for each outcome, with between-group differences slightly attenuated after controlling for demographic variables.

The intervention group demonstrated a significant net weekly decrease in the animal/plant food ratio (−0.03/week, *p* = 0.024) and the intake of livestock and poultry meat (β = −1.80 g/week, *p* = 0.035) compared to the control group. Additionally, the intervention group exhibited a possible weekly increase in the intakes of vegetables and fruits and plant foods with *p* values near the borderline (vegetables and fruits, β = 3.22 g/week, *p* = 0.055; plant foods, β = 3.26 g/week, *p* = 0.057) compared with the control group. In the sensitivity analysis, excluding the participants who changed exercise routine or lunch dietary intake as a proportion of total daily dietary intake during the follow-up period, the intervention effect on reducing the animal/plant food ratio was further confirmed, and the effects on improving intakes of vegetables and fruits and plant foods became significant (animal/plant food ratio, β = −0.03/week, *p* = 0.002; vegetables and fruits, β = 4.41 g/week, *p* = 0.015; plant food, β = 4.52 g/week, *p* = 0.016) (Appendix A).

### 3.4. Effects of the Intervention on Anthropometric Indicators

Table 4 presented the results of two mixed models comparing the overall trends in anthropometric measurements between the two groups monthly from baseline to the end of the study. Compared to the control group, there was a weak declining trend in weight (β = −0.43 kg/month, *p* = 0.074) among the intervention group. The trend became statistically significant in the sensitivity analysis (β = −0.45 kg/month, *p* = 0.004) (Appendix A). Identical to slightly different results on dietary effects from the three models, the inconsistency in results was most likely caused by the combined effect of the small sample size and confounding bias from changes in exercise routine or lunch dietary intake as a proportion of total daily dietary intake during the follow-up period.

### 3.5. Additional Analysis for Ease and Understandability of the Applet

In the post-intervention questionnaire survey, we simply evaluated the ease of use of the applet and the understandability of dietary reports from the applet in the intervention group. The findings indicated that, in the intervention group, a substantial proportion of the participants (94.6%) were able to fully master the applet, while a considerable percentage of the participants (69.6%) could fully comprehend the dietary reports provided by the applet. The remaining participants in the intervention group were able to at least partially utilize the applet or grasp the reports.

## 4. Discussion

The current findings indicated that the applet-based personalized dietary intervention could effectively assist participants in consuming healthier foods, improving the participants’ overall dietary consumption. Furthermore, the participants in the intervention group were more likely to achieve clinically meaningful weight loss. Considering the accessibility and cost-effectiveness of the mobile platform in the real world, these results could be generalized to real-world applications in future dietary intervention programs.

As depicted in Figure 3, the majority of participants exhibited a suboptimal healthy dietary structure at baseline, characterized by inadequate consumption of plant foods and vegetables and fruits, as well as excessive intakes of animal foods and livestock and poultry meat. During the follow-up period, compared to the control group, the intervention group decreased livestock and poultry meat consumption over time, with their vegetables and fruits and plant foods intakes possibly increasing. In China, fruits are commonly used as garnishes for certain dishes rather than being consumed as part of main meals. Therefore, the vegetables and fruits aspect primarily refers to the consumption of vegetables. More importantly, the participants in the intervention group were more likely to hold a deceasing animal/plant food ratio. Thus, for the majority of participants in the intervention group, a balanced and healthy dietary structure was approached through our applet, which referred to meeting China’s Dietary Guidelines [16], including a higher intake of plant foods and a moderate consumption of animal foods. Currently, abundant epidemiological and RCT evidence has found that the improvement of dietary structure can effectively promote participants’ health through several benefits, such as significant weight loss, improved cardiovascular health, and better glucose metabolism [6,23,24]. A large epidemiological study combining databases from 195 countries further indicated that dietary interventions focusing on systematically promoting the multicomponent intakes of diet to the optimal level might have a greater effect than interventions targeting only certain components, such as sugar and fat [1]. The above conclusion was verified in other reviews, which suggested that interventions on only one part of the dietary intakes without controlling the rest would achieve less significant health improvements [25,26]. This is exactly the strength of our study. The applet could provide a real-time and overall display of the participants’ current intakes of various components of a meal along with the corresponding deviations from the recommended intakes. This functionality could effectively and comprehensively assist the participants in making wise and healthy food choices, leading to improved overall dietary intakes.

Corresponding to the aforementioned dietary improvements, a weak weight loss was observed among the participants who had a distribution of BMI and percentage of body fat leaning toward overweight. This was significant in sensitivity analysis. This result suggested that this low-cost intervention under the free-eating condition was most likely to lower weight through changes in dietary intakes. Several studies have demonstrated that approximately 5% reductions in weight with study durations ranging from 6 to 12 months can achieve clinically meaningful prevention of noncommunicable diseases, such as diabetes, cancers, and cardiovascular diseases [27,28]. Based on existing measurement records, we found that the intervention participants lost an average of 0.43 kg of weight monthly on the basis of 66.66 kg (95% CI, 37.39 to 95.92 kg) of baseline weight compared to the control participants. This result made us believe that adherence to the applet for more than six months would more likely help most overweight or obese participants achieve clinically meaningful weight loss.

In contrast to previous dietary intervention studies, our study solely offered the participants real-time individualized dietary information and evaluations per meal within the context of unrestricted eating. Currently, effective dietary intervention projects typically include face-to-face education, dietary limits, assigned recipes, and the provision of healthy foods [7,10,29]. These projects require increased professional involvement and mandatory or partially mandatory dietary changes among participants, resulting in high costs, low compliance, and limited maintenance of effectiveness [30]. In contrast, our intervention solely focused on displaying the participants’ current accurate intakes of various meal components and the respective deviations from the recommended intakes. The participants were not subjected to any dietary restrictions, recommendations, or face-to-face consultations throughout the intervention period. The intervention was solely an effective self-monitoring tool to help the participants visually assess the compositions of food groups and nutrients per meal in order to make healthier food choices. Our findings indicated that this self-monitoring, conducted without dietitians and in a free-eating condition, effectively improved the participants’ dietary intakes and physical conditions, similar to high-investment dietary intervention projects. The results of our study were consistent with previous findings suggesting that self-monitoring is one of the most successful behavioral change techniques for dietary intervention research [9,31]. Based on the self-monitoring, participants would have more awareness of their dietary behaviors and more self-motivation to change these behaviors. Meanwhile, individualized and accurate dietary information per meal was easily available to the participants based on the accessibility and convenience of the intervention provided by the mobile platform. The distinctive superiority is particularly suitable for the fast-growing central kitchen schemas and packaged food industry [32].

For the applet users, the current results showed that our applet was user-friendly and could provide understandable dietary information and evaluations. More importantly, this study was performed in a nonclinical environment that was closer to real-world intervention practices. The participants were encouraged to document their lunches and were given the freedom to stick to their regular daily routines and enjoy meals of their choice, rather than being obligated to follow a prescribed diet. Combining the cost-effectiveness and convenience of digital intervention programs [33], these characteristics would make it possible to generalize the applet in future dietary intervention programs at a national level and to replicate our study results in these programs.

Using the recipe dataset for all the dishes available in the canteen and the Chinese food composition database [12,14], our study could effectively avoid several common defects rooted in traditional dietary survey methods, such as food-frequency questionnaires and three consecutive 24 h recalls [34]. Accurate dietary reports on the intakes of food groups, energy, and nutrients per meal were automatically calculated and provided through the applet. This would avoid potential systematic measurement error due to misreporting personal dietary intake resulting from traditional dietary survey methods, improving the study’s reliability and validity. In addition, these records were produced in real time after meals, which could effectively prevent recall bias.

Figure 4 showed no significant trend in the animal/plant food ratio of the canteen’s daily lunchtime menu after the study began. This suggested that the improvement in the participants’ dietary intake was not attributable to the dining environment. Additionally, this study adjusted results for key confounding factors and demonstrated that the observed associations were robust to these confounders, including age, sex, baseline BMI, smoking status, alcohol consumption, physical activity level, intentional physical exercise, enrollment sequence, and baseline value of the respective dependent variables.

This study has several limitations that need to be acknowledged. First, due to the presence of numerous ready-to-eat meals and mixed sauces in the canteen, precise quantities of cooking oil, sugar, and salt were unavailable, and thus not included in the analysis. However, we collected and analyzed all kinds of nutrients from these components in detail, especially fat, cholesterol, carbohydrate and sodium. Second, this study solely focused on recording and analyzing food and nutrient intake during weekday lunches and provided corresponding dietary reports. Participants’ intake during breakfast and dinner, as well as their dietary habits during the weekends, were not included or monitored. As a result, our findings only partially reflect the intervention effect on participants’ dietary intakes. Following this study, we are designing a new module to record and report dietary intake outside the canteen for participants, enabling the reporting of meals throughout the day and providing dietary guidance. Third, our study did not meet the planned implementation schedule due to the COVID-19 quarantine policy in China, resulting in only two months of the intervention for most of the participants and leaving the follow-up anthropometric measurements for almost half of the participants incomplete (42.5%). As a result, we were unable to observe the long-term effects of the dietary intervention and the effects’ overall trend and maintenance duration. Fourth, the small sample size and short intervention period may limit the validity and generalization of the intervention effect on dietary intakes. Furthermore, the intervention effect on weight loss showed inconsistency between the main analysis and sensitivity analysis due to the small sample size. However, as the first study regarding the applet-based intervention, this study’s results are adequate to determine the feasibility and potential effectiveness of the intervention in real-world scenarios. Additionally, after this study we are conducting a new randomized control trial with a larger sample size, longer intervention period, and standard control to further confirm and generalize the findings of this study. Fifth, evaluating the applet’s understandability and usability based on just two subjective questions may lead to inaccurate conclusions. A more objective method to assess these aspects is warranted. Sixth, the intervention group exhibited a significantly higher mean baseline BMI than the control group, suggesting that there may be notable differences in dietary habits or preferences between the two groups. This disparity could potentially introduce bias in assessing the intervention effects. However, in the main and sensitivity analyses, we have accounted for the baseline BMI value by adjusting them as covariates, mitigating, at least to some extent, the impact of baseline BMI differences on the intervention effects. Finally, the self-reporting of leftover proportions by users might introduce potential deviations from the actual leftover proportions per lunch. Using photographic techniques to record leftover proportions may more accurately approach the actual leftover proportions compared to self-reporting in future studies [35].

## 5. Conclusions

The novel personalized dietary intervention could assist users in improving overall dietary consumption and possibly aid in weight management. This study also suggested that the applet-based dietary intervention was feasible in the real world.

## Figures and Tables

**Figure 1 nutrients-16-00565-f001:**
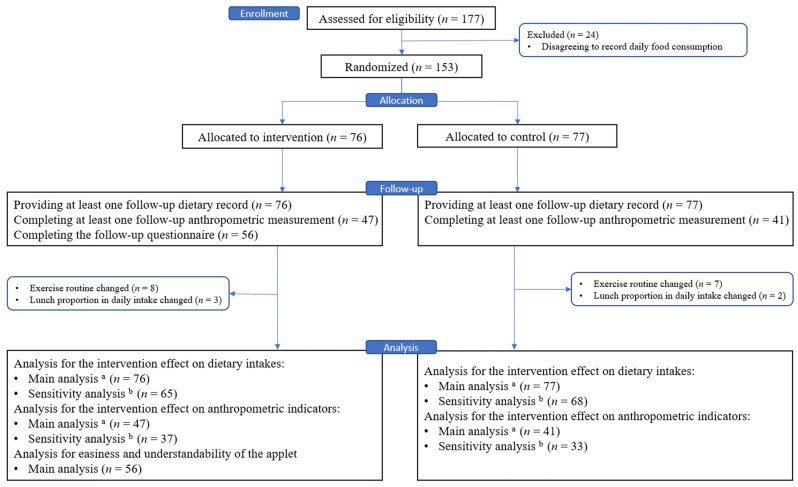
Flow chart of participants. Note: ^a^ analyzed, including the participants who provided at least one diet record or completed at least one anthropometric measurement after the intervention started; ^b^ analyzed, on the basis of the main analysis, further excluding the participants who changed exercise routine or lunch dietary intake as a proportion of total daily dietary intake.

**Figure 2 nutrients-16-00565-f002:**
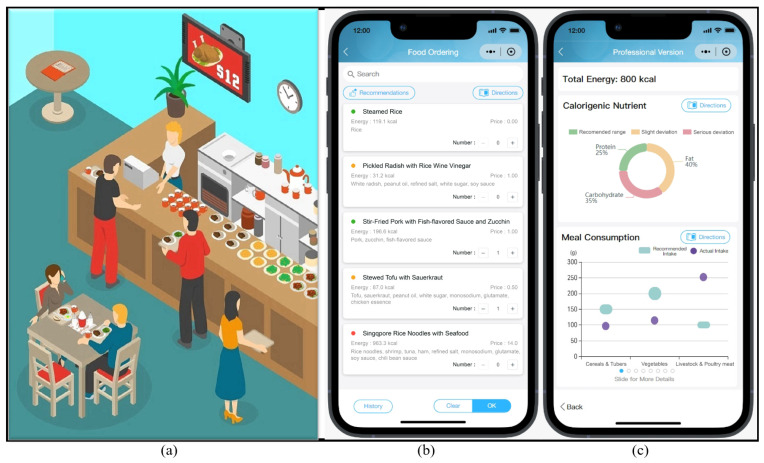
The demonstration of the study canteen and the applet interfaces. Note: (**a**) demonstrates the environment of the study canteen; (**b**) displays the pre-meal dish nutrition evaluation (“traffic light” illustrations) of the applet interfaces; (**c**) displays the post-meal personalized nutrition report of the applet interfaces.

**Figure 3 nutrients-16-00565-f003:**
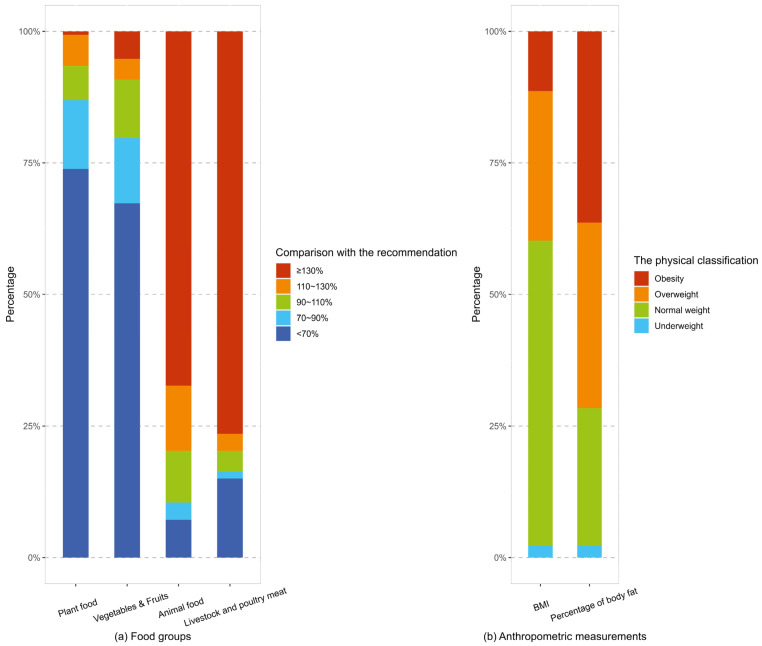
Distributions of baseline lunchtime dietary intakes, BMI and percentage of body fat. Note: Personalized recommended intakes were determined by considering the participants’ sex, age, and EER. Subsequently, the participants’ current intakes were compared to these recommendations and categorized into five levels: <70% of the recommended intake, 70%–90% of the recommended intake, 90%–110% of the recommended intake, 110%–130% of the recommended intake, and ≥130% of the recommended intake. Regarding BMI, the reference standard for the Chinese population divided it into four groups: <18.5 kg/m^2^ (underweight), 18.5 to 24.0 kg/m^2^ (normal weight), 24 to 28 kg/m^2^ (overweight) and ≥28 kg/m^2^ (obesity) [20]. Based on the cutoffs of percentage of body fat proposed by Gallagher [21], and taking into account the higher distribution of percentage of body fat in Asians compared to other populations [22], the percentage of body fat was categorized in a sex-specific manner. In men, the categories were as follows: <10% (underweight), 10% to 20% (normal weight), 20% to 25% (overweight), and ≥25% (obesity). In women, the categories were as follows: <20% (underweight), 20% to 30% (normal weight), 30% to 35% (overweight), and ≥35% (obesity).

**Figure 4 nutrients-16-00565-f004:**
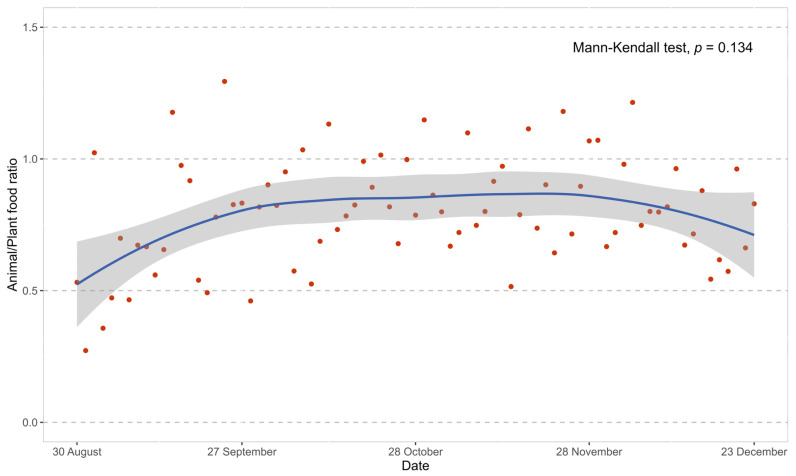
The temporal trend in the animal/plant food ratio of lunchtime food supply. Note: Red dots represented daily animal/plant food ratio, while the smoothing curve displayed the trend of this ratio over time.

**Table 1 nutrients-16-00565-t001:** Cutoffs of the nutrients for dish “traffic light” classification ^a^.

Item (/100 g ^b^)	I	II	III
Fat (g)	<8	8–20	>20
Sodium (mg)	<500	500–1000	>1000
Sugar (g)	<4.5	4.5–9.0	>9.0

Note: ^a^ Green light for the dish represents all three items within the range in the I column, red indicates at least 1 item within the range in the III column, and yellow includes all the others. Additionally, ^b^ 100 g refers to 100 g edible portion of dish.

**Table 2 nutrients-16-00565-t002:** Baseline characteristics of the participants.

	ALL	Control	Intervention	*p* Value
*n*	153	77	76	
Age, years, mean (SD)	32.7 (7.5)	32.7 (7.3)	32.6 (7.8)	0.963
Sex, *n* (%)				
Female	97 (63.4)	51 (66.2)	46 (60.5)	0.572
Male	56 (36.6)	26 (33.8)	30 (39.5)	
Smoking status, *n* (%)				
Non-smoker	136 (88.9)	69 (89.6)	67 (88.2)	0.945
Ex-smoker	11 (7.2)	5 (6.5)	6 (7.9)	
current Smoker	6 (3.9)	3 (3.9)	3 (3.9)	
Alcohol consumption, *n* (%)				
Lifetime abstainer	67 (43.8)	34 (44.2)	33 (43.4)	0.865
Non-heavy drinker	68 (44.4)	35 (45.4)	33 (43.4)	
Heavy drinker	18 (11.8)	8 (10.4)	10 (13.2)	
Daily physical activity level, *n* (%)				
Low	134 (87.6)	70 (90.9)	64 (84.2)	0.341
Moderate	18 (11.8)	7 (9.1)	11 (14.5)	
Vigorous	1 (0.6)	0 (0.0)	1 (1.3)	
Intentional physical exercise, *n* (%)				
No	118 (77.1)	58 (75.3)	60 (78.9)	0.733
Yes	35 (22.9)	19 (24.7)	16 (21.1)	
Enrollment sequence, *n* (%)				
September	23 (15.0)	10 (13.0)	13 (17.1)	0.535
October	86 (56.2)	42 (54.5)	44 (57.9)	
November	44 (28.8)	25 (32.5)	19 (25.0)	
Lunch consumption, mean (SD)				
Food group, g/meal				
Plant foods	222.9 (98.0)	225.9 (94.8)	219.8 (101.6)	0.701
Cereals and Tubers	52.0 (35.6)	51.4 (35.7)	52.63 (35.7)	0.827
Vegetables and Fruits	161.0 (94.7)	163.9 (94.0)	158.0 (96.0)	0.698
Soybeans and soybean products	8.5 (12.4)	8.3 (12.2)	8.7 (12.7)	0.830
Animal foods	126.6 (55.9)	124.8 (51.4)	128.4 (60.4)	0.691
Livestock and poultry meat	84.2 (57.9)	79.4 (56.3)	89.0 (59.5)	0.308
Aquatic products	32.9 (50.3)	35.8 (50.2)	30.0 (50.6)	0.475
Eggs	9.6 (23.6)	9.7 (24.9)	9.5 (22.4)	0.969
Animal/plant food ratio	0.7 (0.6)	0.7 (0.7)	0.7 (0.5)	0.679
Energy, kcal/meal	676.1 (223.7)	666.3 (207.3)	685.9 (240.1)	0.589
Percentage of energy intake from fat	0.5 (0.1)	0.5 (0.1)	0.5(0.1)	0.616
Nutrients				
Protein, g/meal	31.9 (11.8)	31.8 (12.2)	32.0 (11.5)	0.929
Fat, g/meal	35.9 (15.5)	35.1 (14.5)	36.7 (16.4)	0.525
Carbohydrate, g/meal	51.2 (21.8)	51.0 (21.9)	51.3 (21.9)	0.921
Cholesterol, mg/meal	163.4 (168.1)	164.7 (176.0)	162.1 (160.8)	0.925
Sodium, mg/meal	2017.4 (925.1)	1954.2 (916.9)	2081.5 (935.0)	0.396
Calcium, mg/meal	189.6 (119.4)	199.2 (122.5)	179.9 (116.2)	0.321
Iron, mg/meal	6.7 (4.7)	6.3 (2.6)	7.1 (6.1)	0.311
Zinc, mg/meal	5.0 (2.6)	5.0 (2.8)	5.0 (2.5)	0.936
Vitamin C, mg/meal	37.6 (31.6)	39.7 (33.6)	35.6 (29.5)	0.420
Anthropometric measurement, mean (SD)				
Body Weight, kg	64.5 (14.0)	62.9 (14.0)	66.2 (14.0)	0.151
BMI, kg/m^2^	23.2 (3.7)	22.6 (3.8)	23.9 (3.6)	0.033
Blood pressure, mmHg				
Systolic pressure	115.6 (16.1)	114.8 (15.4)	116.3 (16.9)	0.575
Diastolic pressure	75.8 (11.2)	75.2 (10.8)	76.5 (11.6)	0.488
Body composition				
Percentage of body fat, %	28.3 (6.5)	27.8 (6.8)	28.9 (6.1)	0.276
Percentage of torso fat, %	28.9 (6.6)	28.0 (7.0)	29.7 (6.0)	0.112
Visceral fat rank	6.7 (3.8)	6.2 (3.7)	7.2 (3.8)	0.105
Missing, *n* (%)	2 (1.3)	2 (2.6)	0 (0.0)	

Note: BMI, body mass index; SD, standard deviation.

**Table 3 nutrients-16-00565-t003:** Effects of the intervention on average weekly dietary changes of lunch.

	Time × Group ^a^			
	Model 1 ^b^		Model 2 ^c^	
	β ^d^	*p*	β ^d^	*p*
Food group, g/meal				
Plant foods	3.23	0.061	3.26	0.057
Cereals and Tubers	0.08	0.847	0.07	0.870
Vegetables and fruits	3.26	0.054	3.22	0.055
Soybeans and soybean products	0.02	0.933	0.03	0.881
Animal foods	−1.32	0.183	−1.26	0.199
Livestock and poultry meat	−1.75	0.046	−1.80	0.035
Aquatic products	0.60	0.405	0.63	0.378
Eggs	−0.53	0.184	−0.51	0.202
Animal/plant food ratio	−0.03	0.025	−0.03	0.024
Energy, kcal/meal	−3.92	0.143	−4.06	0.130
Percentage of energy intake from fat, %	0.00	0.180	0.00	0.152
Nutrients				
Protein, g/meal	−0.14	0.340	−0.15	0.296
Fat, g/meal	−0.36	0.054	−0.38	0.041
Carbohydrate, g/meal	−0.07	0.746	−0.07	0.748
Cholesterol, mg/meal	−3.87	0.053	−3.97	0.048
Sodium, mg/meal	−3.62	0.765	−4.40	0.718
Calcium, mg/meal	2.29	0.125	2.37	0.115
Iron, mg/meal	−0.03	0.539	−0.04	0.431
Zinc, mg/meal	0.00	0.894	−0.01	0.818
Vitamin C, mg/meal	0.14	0.653	0.03	0.933

Note: ^a^ The significant interaction item (Time × Group) indicated overall trend differences between the both groups; ^b^ Model 1 was adjusted for enrollment sequence and respective outcome variables’ baseline values; ^c^ Model 2 was adjusted for enrollment sequence, the dependent variables’ baseline values, age, sex, baseline BMI, physical activity levels, intentional physical exercise, smoking status and alcohol consumption; ^d^ β value represented average weekly change in dietary intakes.

**Table 4 nutrients-16-00565-t004:** Effects of intervention on average monthly changes of anthropometric indicators.

	Time × Group ^a^			
	Model 1 ^b^		Model 2 ^c^	
	β ^d^	*p*	β ^d^	*p*
Body weight, kg	−0.40	0.099	−0.43	0.074
BMI, kg/m^2^	−0.16	0.142	−0.19	0.091
Blood pressure, mmHg				
Systolic pressure	0.26	0.847	−0.08	0.955
Diastolic pressure	1.38	0.118	1.32	0.135
Body composition				
Percentage of body fat, %	−0.31	0.371	−0.31	0.375
Percentage of torso fat, %	−0.20	0.633	−0.25	0.542
Visceral fat index	−0.16	0.350	−0.14	0.402

Note: ^a^ The significant interaction item (Time × Group) indicated overall trend differences between the two groups; ^b^ Model 1 was adjusted for enrollment sequence and respective outcome variables’ baseline values; ^c^ Model 2 was adjusted for enrollment sequence, the dependent variables’ baseline values, age, sex, physical activity levels, intentional physical exercise, smoking status and alcohol consumption; ^d^ β value represented average monthly change in anthropometric indicators.

## Data Availability

The datasets used and analyzed during the current study are available from the corresponding author upon reasonable request.

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
