# Peer review of "Effects of a Novel Applet-Based Personalized Dietary Intervention on Dietary Intakes: A Randomized Controlled Trial in a Real-World Scenario"

_nutrients, 2024, doi:10.3390/nu16040565_

Round 1

Reviewer 1 Report

Comments and Suggestions for Authors

Dear authors,

This is a really interesting topic and a very well written manuscript. Personalization of nutritional recommendations is nowadays more than crucial and this needs to be addressed. The overall structure and methodology of the study is excellently given.

I only have a few comments:

1. Please add the small follow-up sample size to the limitations' section.

2. Reducing the α level down to 10% is a bit arbitrary. Please further explain its use or at least acknowledge throughout the manuscript the fact that this is arbitrary and not strictly significant 

3. Another limitation is the fact that the understandability and usibility of the applet was subjectively assessed by asking the participants. A more objective method of assessing it would be more accurate to further assume the generalizability of your results.

4. Line 125: Chinese

Author Response

Thank you very much for taking the time to review this manuscript entitled “Effects of a Novel Applet-Based Personalized Dietary Intervention on Dietary Intakes: A Randomized Controlled Trial in a Real-World Scenario” (Manuscript ID: nutrients-2863567). According to your valuable suggestions, we have revised our manuscript and have added new information where appropriate.

The point-by-point response has been sent as the email attachment. Please see the attachment.

Reviewer 2 Report

Comments and Suggestions for Authors

The authors present a randomized controlled study on the effects of a novel applet-based dietary intervention. The manuscript is well-written, clear, easy-to-understand. The English of the manuscript is correct. The primary outcome parameter was the change in dietary intake and the secondary outcome parameters were the anthropometric changes. Obviously slight changes were expected and obtained in the primary outcome parameters.

1/ I suggest to call the results preliminary due to the low number of participants, especially low number in the anthropometric part (47 in the intervention and 41 in the control group).

2/The low number of observations does not allow the authors to analyse gender differences. This is the case with the effect of body composition (normal, overweight, obese) at baseline on the effectivity of the intervention. 3/The BMI at baseline was significantly different of the two groups. This should be discussed.

The study is interesting and deserves publication.

Author Response

(The authors gave the same response as above.)

Reviewer 3 Report

Comments and Suggestions for Authors

Comments for the Authors:

             Dr. Liu and colleagues have conducted a 4 month RCT in a relatively small group to test whether their digital device whose description in the title is unclear could shift lunch time intake.  It appears to have done that, although the clinical significance and statistical significance are “weak”.   As the authors no doubt know there are efforts to use photographic data collected before and after a meal to provide a more accurate representation of what is eaten and not eaten.  Because there is not intermediary other than the camera, this seems a more viable long-term strategy. 

Specific Comments:

Title:  The title could be clearer.  By Applet are the authors speaking of a hand held computer or something else.  Try to make the Title more specific.

Abstract: 

Line 20:          This reviewer has not idea what an “applet” is.  Please use a more generic term.

Introduction: 

Line 36:          Unhealthy might be a better word that “subhealth”

Line 56:          What is an “applet”?

Methods:

Line 64:          The use of the Applet appears to require quite restricted ways of preparing and displaying foods.  Is that correct?  How to you handle the “real world” where these restrictions don’t exist.

Line 117:        Were all of the participants eating in the same venue?   And at the same time?  How did you keep people from sharing information?

Line 156:        How was the Estimated Energy Requirement calculated?

Line 175:        Was this trial done only at lunchtime?

Line 194:        Left overs can pose a real problem  How were they assessed?

Line 209:        Delete “s” at end of averages

Line 230:        Delete “d” from examined.

Results:

Line 305:        Figure 4 and your description seem to say opposite things.  In the figure the animal/plant ratio increases, but in the text you say “net weekly decrease in the animal/plant food ratio (-0.03 /week, p=0.024)”  Please reconcile.

Line 328:        Your value of 0.074 isn’t even close to the value you set for significance.

Line 345:        Please rephrase “comprehensively comprehend” without using the same word twice.

Discussion:

Line 351:        Under conditions which are quite unlikely to exist in the real world.

Line 352:       In Table 4 the weight loss is not statistically significant nor clinically  meaningful in his reviewers mind. 

Line 354:        You have not convinced me that it could be made useful in as you say, the “real world”.

Line 385:        “Free living” is stretching it in this reviewer’s opinion.  The conditions to this reviewer appear to be quite constrained.

Line 394:        This is a contention that must be tested.  Most weight changes after interventions occur in the short term, no the long term.

Line 402:        The accuracy of the input depends on the knowledge of composition of the dishes chosen, and the accuracy of reporting “left overs”, a serious problem for accuracy.

Line 432:        There is no question that digital technology can improve data collection, and calculation, but it is limited in the current study by the information about the foods eaten and the report of uneaten portions.  Use of photographic techniques for recording initial and final meals, seems to this reviewer a much better strategy to pursue.

Line 437:        As noted above your figures and text seem to go in opposition directions.

Figures and Tables:  Good as presented.

Comments on the Quality of English Language

Well written and deserves acceptance after minor revisions.

Author Response

(The authors gave the same response as above.)
